# Anti-Diabetic Activity of Glycyrrhetinic Acid Derivatives FC-114 and FC-122: Scale-Up, In Silico, In Vitro, and In Vivo Studies

**DOI:** 10.3390/ijms241612812

**Published:** 2023-08-15

**Authors:** Samuel Álvarez-Almazán, Luz Cassandra Solís-Domínguez, Paulina Duperou-Luna, Teresa Fuerte-Gómez, Martin González-Andrade, María E. Aranda-Barradas, Juan Francisco Palacios-Espinosa, Jaime Pérez-Villanueva, Félix Matadamas-Martínez, Susana Patricia Miranda-Castro, Crisóforo Mercado-Márquez, Francisco Cortés-Benítez

**Affiliations:** 1Laboratory of Biotechnology, Unidad de Posgrado, Facultad de Estudios Superiores Cuautitlán Campus 1, Universidad Nacional Autónoma de México, Cuautitlán Izcalli 54740, Mexico; samuel.alvarez@cuautitlan.unam.mx (S.Á.-A.); cassandrasolis433@gmail.com (L.C.S.-D.); teresafuerte52@gmail.com (T.F.-G.); mariaaranda500@gmail.com (M.E.A.-B.); spmcunam55@gmail.com (S.P.M.-C.); 2Laboratory of Synthesis and Isolation of Bioactive Substances, Departamento de Sistemas Biológicos, División de Ciencias Biológicas y de la Salud, Universidad Autónoma Metropolitana–Xochimilco (UAM–X), Mexico City 04960, Mexico; duper_pau@icloud.com (P.D.-L.); jpalacios@correo.xoc.uam.mx (J.F.P.-E.); jpvillanueva@correo.xoc.uam.mx (J.P.-V.); felixmatadamas@yahoo.com.mx (F.M.-M.); 3Laboratory of Biosensors and Molecular Modelling, Departamento de Bioquímica, Facultad de Medicina, Universidad Nacional Autónoma de México, Mexico City 04510, Mexico; martin@bq.unam.mx; 4Isolation and Animal Facility Unit, Facultad de Estudios Superiores Cuautitlán 28, Universidad Nacional Autónoma de México, Cuautitlán Izcalli 54714, Mexico; crisofo@cuautitlan.unam.mx

**Keywords:** glycyrrhetinic acid, type 2 diabetes mellitus, semisynthesis, molecular docking, molecular dynamics, protein tyrosine phosphatase 1B, α-glucosidase, acute oral toxicity

## Abstract

Type 2 diabetes (T2D) is one of the most common diseases and the 8th leading cause of death worldwide. Individuals with T2D are at risk for several health complications that reduce their life expectancy and quality of life. Although several drugs for treating T2D are currently available, many of them have reported side effects ranging from mild to severe. In this work, we present the synthesis in a gram-scale as well as the in silico and in vitro activity of two semisynthetic glycyrrhetinic acid (GA) derivatives (namely FC-114 and FC-122) against Protein Tyrosine Phosphatase 1B (PTP1B) and α-glucosidase enzymes. Furthermore, the in vitro cytotoxicity assay on Human Foreskin fibroblast and the in vivo acute oral toxicity was also conducted. The anti-diabetic activity was determined in streptozotocin-induced diabetic rats after oral administration with FC-114 or FC-122. Results showed that both GA derivatives have potent PTP1B inhibitory activity being FC-122, a dual PTP1B/α-glucosidase inhibitor that could increase insulin sensitivity and reduce intestinal glucose absorption. Molecular docking, molecular dynamics, and enzymatic kinetics studies revealed the inhibition mechanism of FC-122 against α-glucosidase. Both GA derivatives were safe and showed better anti-diabetic activity in vivo than the reference drug acarbose. Moreover, FC-114 improves insulin levels while decreasing LDL and total cholesterol levels without decreasing HDL cholesterol.

## 1. Introduction

Diabetes is a complex, chronic, and progressive metabolic disease derived from insulin absence or low activity that ends in hyperglycemia (when the fasting plasma glucose level is greater than or equal to 126 mg/dL) [1]. These abnormally high plasma glucose levels usually lead to major complications, including coronary heart disease, chronic kidney disease, peripheral neuropathy, peripheral vascular disease, oral disorders, and retinopathy [2,3,4]. In addition, other serious complications have been associated with diabetes, such as an increased risk of developing cancer, liver disease, infection-related complications, and cognitive and affective disorders [5]. Diabetes is a common and increasingly prevalent disease of significant public health concern, with 536.6 million diagnosed cases worldwide in 2021, expected to increase to 783.2 million by 2045 [6]. Diabetes has a higher prevalence and mortality in low- and middle-income countries [6,7]. For instance, in Mexico, around 14.1 million people (11%) were diagnosed with diabetes in 2021 [1], and the disease was responsible for an average of 101,496 deaths during 2017–2019, rising dramatically to 148,437 (46%) in 2020 due to the onset of the COVID-19 pandemic [8].

Of the three main types of diabetes (type 1, 2, and gestational), type 2 diabetes (T2D), or non-insulin dependent diabetes mellitus, is the most common, comprising 90–95% of all the cases [9]. The T2D course is characterized by a decline of beta cell function as well as the decreased activity of the phosphatidylinositol-3-kinase (PI3K) and protein kinase B (PKB) pathways [10]. This is mainly due to an altered insulin signaling pathway, resulting in the decreased activity of glucose transporters (GLUT), mainly GLUT1 and GLUT4 [11,12]. Consequently, glucose cannot cross the plasma membrane on muscle, liver, and adipose tissue [13,14]. This phenomenon is also known as insulin resistance due to the body’s inability to use insulin effectively [10]. 

Despite the complex pathogenesis of T2D, numerous molecular pathways and targets have been elucidated to develop novel drugs for its treatment [15,16,17]. Among these, the major types of drugs have been described as secretagogues (including sulfonylureas and glinides such as glibenclamide and repaglinide, respectively), insulin mimickers (DPP-4 inhibitors or gliptins such as sitagliptin and linagliptin) and insulin sensitizers (PPAR-ɣ agonists such as pioglitazone (PIO) and rosiglitazone), starch blockers (α-glucosidase inhibitors such as acarbose and voglibose), and renal glucose reabsorption reducers (SGLT2 inhibitors such as canagliflozin and dapagliflozin). Although these drugs have proven effective in controlling blood glucose levels and managing diabetes, there are many reports in which they present several side effects that range from mild to severe, including digestive disturbances (such as nausea and diarrhea), anemia, neuropathy, hypoglycemia, cardiovascular risk, and bladder cancer [18,19,20,21]. Therefore, developing safe and effective drugs for controlling T2D that reverse the course of its complications is remarkably necessary.

Natural products have become an essential source of bioactive agents for discovering new drugs to treat T2D [22,23,24,25]. In this sense, glycyrrhizic acid (GL, a glycosylated pentacyclic triterpene extracted abundantly from the licorice roots of the plant *Glycyrrhiza glabra*) [26,27,28] and its triterpene aglycone glycyrrhetinic acid (GA) (Figure 1) have demonstrated a remarkable in vivo anti-diabetic activity in rodents [29]. For instance, an oral dose of 100 mg/kg of GA improved blood glucose and insulin levels compared with glibenclamide (600 μg/kg/body weight) in streptozotocin-induced diabetic rats [30,31]. Additionally, treatment with GA at the same dose restored normal levels of triacylglycerols (TAG), total cholesterol, and their fractions: high-density lipoprotein (HDL), very low-density lipoprotein (VLDL), and low-density lipoprotein (LDL) in the plasma of diabetic rats [32].

It has been described that GA has the ability to interact weakly with multiple target proteins, including Protein Tyrosine Phosphatase 1B (PTP1B) and α-glucosidase enzymes [33,34]. PTP1B is a central regulator of glucose homeostasis and energy expenditure of its crucial role in the negative regulation of insulin and leptin signaling pathways [35,36,37]. High levels of PTP1B protein, particularly in the hypothalamus, are associated with insulin and leptin resistance [37]. PTP1B knockout mice have shown improved glucose homeostasis, reduced weight gain, and lower energy expenditure [38,39]. PTP1B inhibitors can serve as insulin mimickers, as well as insulin and leptin sensitizing agents [40], therefore making them an attractive option to combat T2D and obesity simultaneously, which represents an advantage over existing therapies to date. However, developing PTP1B inhibitors remains challenging because they often possess a highly negatively charged group to interact at the catalytic site, which impairs their pharmacokinetic properties. Similarly, the highly conserved catalytic domain of the PTP’s family promotes these inhibitors to be less selective, thus increasing their toxicity [41,42,43,44]. Hence, currently, there are no approved anti-diabetic drugs that target PTP1B activity. On the other hand, α-glucosidase is an exoenzyme located in the brush border of the small intestine [45]. This enzyme hydrolyzes complex carbohydrates into simple ones by means of cleavage α (1→4) bonds linked to α-D-glucose. The released glucose is then absorbed into the intestine and passes to the bloodstream. α-glucosidase inhibitors (AGIs) are oral drugs widely used for T2D treatment alone or combined with other anti-diabetic drugs. They inhibit the absorption of carbohydrates from the small intestine [46]. The AGIs are beneficial in reducing postprandial glucose levels and for patients at risk of hypoglycemia or lactic acidosis [47,48,49]. Some of the most commonly used AGI medications on the market include voglibose (VOG), miglitol, and the widely prescribed acarbose [50,51,52]. 

Since both PTP1B and α-glucosidase are very attractive molecular targets for the development of anti-diabetic drugs and GA can inhibit both enzymes, we consider that structural modifications of the GA scaffold can obtain potent dual PTP1B/α-glucosidase inhibitors. In this sense, we previously reported the synthesis and PTP1B inhibitory activity of indole- and *N*-phenylpyrazole-GA derivatives namely FC-114 and FC-122 (Figure 1) [53]. These semisynthetic compounds were found to be 20- to 30-fold more potent than GA. Moreover, they demonstrated more potency over positive controls: ursolic acid, claramine, and suramin as PTP1B inhibitors. However, no reported in vitro experiments have determined whether FC-114 and FC-122 are α-glucosidase inhibitors or have anti-diabetic potential. To address this, we conducted in vitro tests on the compounds against α-glucosidase and used molecular docking and molecular dynamic (MD) studies to investigate the hypothetical binding mode of FC-122 in α-glucosidase. Furthermore, we examined the compound’s ability to improve glucose, insulin, TAG, and cholesterol (HDL, calculated LDL, and total) levels in a streptozotocin-induced diabetic rat model. We also assessed their in vitro cytotoxicity on Human Foreskin fibroblasts and in vivo acute oral toxicity (AOT) studies. Lastly, we compared the anti-hyperglycemic activity of streptozotocin-induced diabetic rats treated with FC-114 and FC-122 to reference drugs glibenclamide, PIO, and acarbose (Figure 1).

## 2. Results and Discussion 

### 2.1. Chemistry

Unlike our previous work in which FC-114 and FC-122 were prepared on a milligram-scale from GA [53], herein we synthesize both compounds on a gram-scale from GL, which is readily available in natural root plant extracts of *Glycyrrhiza glabra* (Figure 1). Briefly, GL was hydrolyzed by aqueous HCl to obtain GA. Intermediates 3-oxo-GA (2) and 2-formyl-3-oxo-GA (3) were prepared using Jhones oxidation and Claisen condensation, respectively. We found that using tetrahydrofuran (THF) as a solvent for both reactions resulted in better yields and shorter reaction times compared to those using acetone or dioxane, respectively. Finally, FC-114 was prepared from compound two by Fischer indolization using 4-(trifluoromethyl)phenylhydrazine hydrochloride in refluxing acetic acid (AcOH), while FC-122 was synthetized by treating compound three with 4-tolylhydrazine hydrochloride in dry ethanol (EtOH). The overall yield of FC-114 and FC-122 starting from GL was 67% and 15%, respectively. 

### 2.2. Enzymatic Kinetics of α-Glucosidase and PTP1B Inhibition

Since both FC-114 and FC-122 share the GA skeleton and this structure has demonstrated α-glucosidase inhibitory activity, we tested both analogs against α-glucosidase from *Rominococcus obeum* at fixed inhibitor concentrations varying the substrate (*p*-nitrophenyl glucopyranoside, pNPG) concentrations according to the procedure reported elsewhere [54]. Results showed that FC-114 was not active in the range of concentrations tested, while FC-122 was found to be ~5-fold better potency than that shown by acarbose with K_i_ values of 5.2 and 26.8 µM, respectively. Figure 2A,B discloses the Lineweaver-Burk plots showing the effect of acarbose and FC-122 on the α-glucosidase catalytic activity. For acarbose, the lines with the same y-intercept but different slopes confirm its competitive inhibitory activity, as reported in numerous studies [55,56,57,58]. Conversely, it can be seen in Figure 2B that the inhibition mode of FC-122 was of linear-mixed type.

As pointed out in this paper’s introduction, we have reported the PTP1B inhibitory activity of FC-114 and FC-122 [53]. This study elucidated the non-competitive inhibition (K_i_ = 3.9 µM) of FC-114 against PTP1B. Nevertheless, the inhibition type of FC-122 remained unknown to date. Therefore, we performed an enzyme kinetic assay for PTP1B_1–400_ [59,60,61] using *p*-nitrophenyl phosphate (pNPP) as substrate at different concentrations as well as the inhibitor. Figure 2C depicts the double reciprocal plot of Lineweaver-Burk, showing the effect of FC-122 on the PTP1B activity. The parallel lines mean FC-122 behaves, in this case, as an uncompetitive inhibitor with a K_i_ value of 0.44 µM. Since the selective inhibition of PTP1B, among other PTPs with high homology (such as T-cell PTP) is desirable, the uncompetitive inhibition of PTP1B by FC-122 represents a promising strategy for developing a novel class of PTP1B inhibitors because, as is well known, uncompetitive inhibitors do not interact at the catalytic site domain (which is highly conserved among the PTP’s family), but instead bind to an allosteric site of enzymes only when the substrate is already bound to the catalytic site (E-S complex), thus forming an Enzyme-Substrate-Inhibitor complex (E-S-I complex) [62,63].

Considering these results, the addition of an *N*-(4-methylphenyl)pyrazole moiety to the GA skeleton resulted in a potent dual PTP1B/α-glucosidase inhibitor with an uncompetitive and mixed-type inhibitory activity, respectively. This is an interesting finding from the polypharmacological perspective, as the compound FC-122 can interact with two targets related to T2D. Thus, in theory, this compound can increase insulin sensitivity and reduce the intestinal absorption of glucose, making it a recommended first-line approach to reduce the risk of developing T2D [64].

### 2.3. Molecular Docking 

To investigate the potential binding modes of FC-122 into an α-glucosidase enzyme, we carried out docking simulations employing CB-Dock2 server [65,66], Autodock (AD) (The Scripps Research Institute, La Jolla, CA, USA) [67] and GOLD (The Cambridge Crystallographic Data Centre, Cambridge, UK) version 2022.3.0 [68] programs as well as the tridimensional structure of α-glucosidase from *R. obeum* (PDB ID: 6c9z). For comparison purposes, the structure of acarbose was also docked. Since docking validation is an important aspect of measuring the docking protocol’s quality, we re-docked the co-crystal structure of the VOG found in the 6c9z complex. To identify the preferred binding sites of FC-122, we performed blind docking using the CB-Dock2 server. Afterward, site-specific docking was used at its preferred binding site using AD and GOLD. Conversely, because acarbose and VOG are competitive α-glucosidase inhibitors, both compounds were docked at the catalytic binding site using the programs above.

As shown in Table 1, the docking study provided low binding energy in AD as well as high ChemPLP fitness score (CFS) and GOLD score (GS) values in GOLD for the best docking conformation of VOG, indicating a favorable binding at the catalytic binding site. Moreover, after superposing the binding pose of the re-docked structure onto the native pose of VOG, a root mean square deviation (RMSD) value less than 1.5 Å was obtained, depicting the accuracy of both programs (Appendix A). Acarbose fitted at the catalytic binding site with better binding energy and CFS values than VOG, suggesting better binding. This inhibitor formed nine hydrogen bonds within the catalytic binding site of subunit A, including Asp^73^, Asp^197^ (two H-bonds), Arg^404^ (two H-bonds), Asp^420^, Lys^422^, and His^478^ (Figure 3). In addition, the positively charged amine group of acarbose performs fundamental salt bridge interactions with Asp^73^ and Asp^420^. It is particularly worth mentioning that we found very similar interactions (H-bonds and salt bridge interactions) in the crystal structure of Human Maltase-Glucoamylase (MGAM) in complex with acarbose (PDB ID: 2QMJ). MGAM [69] is also an enzyme responsible for catalyzing the last glucose-releasing step in starch digestion and shares a high homology with α-glucosidase from *R. obeum*. The blind docking simulation using the CB-Dock2 server revealed that FC-122 preferentially binds into a hydrophobic pocket close to subunit A’s catalytic domain. The binding energy of FC-122 was similar to acarbose, while the GS value of this compound was more favorable, indicating better binding. The triterpene skeleton of FC-122 formed hydrophobic interactions with Pro^75^, Ile^76^, Phe^314^, and Val^351^. Furthermore, the *N*-phenylpyrazole moiety interacts through π-π stacked and π-π T-shaped interactions with Trp^271^ of subunit A and Trp^341^ within subunit B. In contrast, the methyl group at the phenyl moiety formed a hydrophobic contact with Trp^271^ of subunit A. Finally, the carboxylate group at the C-30 position of FC-122 interacted through a salt bridge contact with the side chain of Lys^348^ within subunit A.

### 2.4. Molecular Dynamics Simulations

We performed 200 ns Molecular dynamics (MD) simulations for the α-glucosidase protein system and α-glucosidase protein-ligand complexes obtained from the docking simulations. The MD studies employed the YASARA structure software (version 22.9.24) [70] and AMBER14 forcefield [71]. To make the RMSD and root-mean-square-fluctuation (RMSF) plots, we analyze the MD trajectories by running the *md_analyze* macro. Furthermore, Molecular Mechanics Poisson-Boltzmann Surface Area (MM-PBSA) calculations were performed to calculate the binding energy of each ligand onto the α-glucosidase during the entire MD simulation. 

Results of the RMSD analysis for the overall fluctuation of the α-glucosidase’s Cα carbons (Figure 4A) showed that the α-glucosidase system tends to stabilize after 120 ns with an average RMSD value of 1.74 Å. Interestingly, the predicted RMSD value for the α-glucosidase-acarbose system was 1.96 Å, whereas the α-glucosidase-FC-122 system gave a more stable complex with a lower average RMSD value of 1.74 Å. Moreover, the RMSD values of this complex were lower from 105 ns until the end of the MD simulation than that showed by the α-glucosidase system, indicating that FC-122 can stabilize the overall conformation of α-glucosidase. In addition, the fluctuation of amino acid residues due to ligand binding was measured by the RMSF values. Figure 4B depicts the RMSF versus residue number. From the chart, the α-glucosidase-acarbose complex showed lower fluctuations in the amino acid sequence located within the catalytic-binding site of subunit A (residues Asp^73^, Asp^307^, Arg^404^, and Trp^417^) when compared to the α-glucosidase protein system. However, higher values of RMSF in the same sequence were observed for the α-glucosidase-FC-122 complex, particularly in residues Asp^73^ and Asp^420^, indicating a significate conformational change in the catalytic region. These residues play a key role in the hydrolytic cleavage of oligosaccharides, particularly the side chain of Asp^420^. Thus, a conformational change within this region may affect the oligosaccharides’ substrate recognition and turnover.

On the other hand, after MD simulations, the position of both inhibitors slightly changed (Figure 4C). For instance, the average movement of acarbose and FC-122 (Figure 5) after superposing on the protein structure was 4.2 and 5.9 Å, respectively, suggesting that both compounds performed stable interactions at their binding sites during the simulation time. Indeed, based on the analysis of the two ligand-contact graphs (Appendix A), acarbose formed new hydrophobic interactions with residues Pro^75^, Tyr^169^, Ile^198^, Trp^271^, Met^308^, Phe^314^, and Phe^453^, while maintaining for a short time the salt bridge interaction with Asp^420^ as well as the H-Bond interactions with Tyr^169^, Asp^197^, Lys^348^, Arg^404^ and Lys^422^. Acarbose formed stable H-bond interactions with residues Asp^73^ and Asp^307^ during the MD simulation. Conversely, it was found that FC-122 moved slightly from its interaction site to produce new hydrophobic interactions with residues from both subunit A (Ile^198^, Lys^348^, and Phe^453^) and subunit B (Trp^341^, Gln^344^, and Ala^345^) while maintaining the hydrophobic and pi-stacked interactions with residues of subunit A (Pro^75^, Ile^76^, and Phe^314^) and subunit B (Trp^341^). Furthermore, FC-122 lost the salt bridge interaction with Lys^348^ of subunit A (which became a hydrophobic contact), but it formed a new salt bridge interaction between its carboxylate group and the positively charged amine group of Lys^348^ within subunit B. To quantitatively address the affinity of the two inhibitors, we performed a binding energy analysis as a function of the simulation time for 200 ns using the MM-PBSA method (Figure 4D). The binding energy was obtained by calculating the energy of the ligand-protein complex system (i.e., the bound state) and subtracting the energy at an infinite distance between the ligand and the rest of the protein system (i.e., the unbound state) from every 100 ps. More negative values indicate better binding in the context the of AMBER14 forcefield. FC-122 demonstrated the best binding energy profile (average = −44.6 kcal/mol) compared to acarbose (average = 40.0 kcal/mol) during the MD simulation. The MD results suggest that FC-122 forms a stable complex with the α-glucosidase enzyme and binds strongly to the catalytic domain’s adjacent region (allosteric site). We believe that this binding may distort the structure of the catalytic pocket so that the substrate molecule does not fit as snugly as before. As a result, less substrate is turned over. The data of molecular docking and MD simulations are in accordance with our enzymatic kinetic studies where FC-122 is a mixed-type α-glucosidase inhibitor.

### 2.5. Cytotoxic Assay and Acute Oral Toxicity

GA and GL are commonly used as flavoring and cosmetic ingredients. However, at high concentrations, they can be harmful to cells. For humans, doses less than 100 mg/day are considered safe [72]. Since FC-114 and FC-122 have similar structures to GA and are potential drugs, their toxicity needs to be evaluated alongside their inhibition potential. Therefore, we conducted cytotoxicity assays on Human Foreskin fibroblasts (HFF-1) and an AOT on female Wistar rats to assess their safety. The concentration that reduced the proliferation of HFF-1 by 50% (CC_50_) was determined for compounds FC-122, FC-114, and GA (Figure 6). Our findings revealed that the structural modifications made to the GA skeleton by adding either an indole or *N*-phenylpyrazole ring resulted in similar low cytotoxicity values compared to GA. The *N*-phenylpyrazole derivative was slightly better than the indole one. Among the derivatives tested, FC-122 had the least cytotoxicity, with a CC_50_ value of 68.4 ± 1 µM. GA and FC-114 had similar cytotoxic effects against HFF-1 cells, with CC_50_ values of 62.0 ± 2 µM and 59.5 ± 3 µM, respectively. These results suggest that FC-114 and FC-122 have a comparable cytotoxic profile to GA and may be harmful at high concentrations.

We then conducted a study to determine the potential toxicity of newly developed compounds in female Wistar rats. After administering FC-144 and FC-122 at three varying doses, our findings showed that there were no toxic effects on the rats (Table 2, Figure 7 and Figure 8). The AOT of these compounds is higher than 2000 mg/kg. It is worth noting that both FC-114 and FC-122 have a much lower AOT than the lethal dose of 50% (LD_50_) reported by GA (LD_50_ = 610 mg/kg) [72]. Therefore, these results indicate that 52 mg/kg or less could be administered safely in normal and streptozotocin-induced diabetic rats.

### 2.6. In Vivo Administration of FC-114 and FC-122 in a Wistar Rat Diabetes Model

It has been observed that STZ can produce a rat model with severe weight loss [32], signs, and symptoms [13,73,74,75,76,77] similar to those of diabetic patients. The extent of weight loss is proportional to the dosage [73,74,75,76]. Furthermore, insulin typically has an anabolic effect on protein metabolism by promoting protein synthesis and slowing protein degradation [78]. However, administering 45 mg/kg of STZ (in week 1) did not result in weight loss in the rat model (as shown in Figure 9A). Subsequently, treatments with FC-114 (*p* > 0.01) and PIO (*p* > 0.001) (administered for 14 days in weeks 5 to 7) resulted in a difference in weight when compared to the untreated group (T2D). Overall, all groups displayed similar behavior during the two weeks of treatment (weeks 5 to 7), suggesting that the treatments did not lead to significant weight gain or maintenance. While chronic use of PIO has been linked to weight gain in previous studies [52,79,80], this was not observed during the initial days of treatment [81]. This may be due to the patient’s homeostasis being achieved, leading to decreased hunger and reduced weekly feeding. The same trend was observed for the other treatments (including FC-114 and FC-122), except for glibenclamide treatment.

The consumption of food is related to weekly weight changes. The results of this study indicate that all the treatments led to a decrease in food consumption compared to untreated animals, as shown in Figure 9B. Interestingly, after 7 days of treatment (week 6), FC-122 reduced food intake to near-normal levels. However, diabetic animals treated with FC-114 maintained their increased food consumption during the first week of treatment (weeks 5 to 6). After 14 days of treatment (weeks 6 to 7), their food intake tended to decrease toward normal levels. These results of decreased food intake displayed by FC-114 and FC-122 are promising because animals with T2D (like patients) tend to consume more food and water, which is known as polyphagia and polydipsia, respectively [1,13]. Therefore, compounds like FC-114 and FC-122 may help reduce appetite and daily caloric intake in patients with T2D. 

Untreated diabetic animals had a constant water consumption, while all treatments caused a decreasing trend after 14 days of administration (Figure 9C). The compound FC-122 showed a decreased water consumption compared to acarbose and FC-114. Treatments that reduce hyperglycemia levels in diabetic patients/animals also decrease fasting glucose, which in turn produces molecular variations such as less thirst (due to the osmotic capacity of glucose) and less presence of the hunger stimulus (due to insulin sensitization and the ability of glucose to enter tissues sensitive to this hormone) [10,15,82,83,84]. Therefore, there is no feedback to generate the hunger stimulus, and the animals stop eating a lot.

The fasting glucose test is a commonly used method for diagnosing diabetes [1,82,83]. The results of this study indicate that there were differences observed between the effects of acarbose and FC-114 (*p* > 0.01), FC-122 (*p* > 0.01), and PIO (*p* > 0.0001) (Figure 10A). Diabetes is a disease characterized by blood glucose levels above 126 mg/dL [1], and these levels tend to rise over time [1,84]. While the treatments administered did not produce a significant difference in diabetic animals, it was observed that glibenclamide and acarbose showed a downward trend, whereas FC-114 and FC-122 were able to halt the continuous increase of glucose levels after 7 and 14 days of treatment (during weeks 6 and 7).

Before being given a single dosage of 45 mg/kg, the STZ-generated diabetes model had inconsistent fasting glucose results and glucose tolerance curves at the 5-week mark. However, by the end of week 5, the results become more uniform [13]. We decided to start administering treatments at this point to compare their effectiveness. All treatments were given at the same equimolar dose of PIO, as previous research showed that 30 mg/kg of this drug could lower blood glucose and TAG l levels, as well as the area under the curve (AUC) during the oral glucose tolerance test (OGTT) after 14 days of administration [13]. It was important not to use a higher dose that could cause euglycemia (normal glucose levels) so that we could accurately compare the efficacy of the different treatments.

Based on Figure 10, it can be observed that the OGTT results indicated a decrease in glycemia values after 7 and 14 days of treatment with the anti-diabetic drugs, FC-114, and FC-122 (weeks 6 and 7, Figure 10C,D) compared to day 1 of treatment (week 5, Figure 10B) (except to FC-114 and glibenclamide). The most significant decrease in glycemia was observed with PIO at 14 days of treatment (week 7). In terms of effectiveness (at week 7) in lowering blood glucose values during the glucose tolerance curve, and therefore AUC (Table 3), PIO was the most effective, followed by glibenclamide, FC-122, FC-114, and acarbose.

After conducting in vivo tests and statistics analysis using ANOVA and Bonferroni multiple comparison test, it was found that 30 mg/kg of PIO significantly improved glucose blood levels in OGTT after 14 days of administration (Figure 10D) compared to untreated animals (*p* < 0.0001). Glibenclamide treatment (*p* < 0.01) ranked second, while FC-122 (*p* < 0.05) ranked third. There were differences between treatments: PIO and acarbose (*p* < 0.00001), FC-114 (*p* < 0.0001), FC-122 (*p* < 0.001), and glibenclamide (*p* < 0.01). Acarbose, FC-122 (*p* < 0.0001), and FC-114 (*p* < 0.001) were also different, with FC-122 being found to be the best anti-hyperglycemic agent among the three compounds tested. The contribution of blood glucose from the diet is very important, so the inhibitory activity of acarbose on glucosidases (including salivary and pancreatic) considerably improves blood glucose levels [52]. FC-122 being a better α-glucosidase inhibitor and inhibiting PTP1B could explain its better ability to lower high glucose levels than FC-114. This suggests that a dual PTP1B/α-glucosidase inhibitor may result in better effectiveness. However, it should be noted that the compounds were found in more significant amounts in the gastrointestinal tract compared to the anti-diabetic drugs, indicating that they were poorly absorbed and reached target tissues (where PTP1B is ubiquitously expressed) at small concentrations. Therefore, the anti-hyperglycemic activity of FC-122 can be attributed to its inhibition of α-glucosidase, an enzyme located on the surface of the intestinal epithelial cells [45]. It is important to highlight that the doses of the commercial treatments and the compounds tested are equimolar to 30 mg/kg of PIO, so observing these subtle differences provides the guideline to continue exploring the most appropriate doses of the compounds, avoiding falling into non-secure dose (toxic). For example, the effective doses to cause decreases in blood glucose are 2.5–20 mg/kg for glibenclamide [80], 15–45 mg/kg for PIO [80], 100 mg/kg for GA (which is structurally related to FC-122 and FC-114) [53], and 75–300 mg/kg for acarbose [80].

The biochemical parameters tested in serum showed no significant differences between the treatments (Figure 11), although there was a slight tendency to reduce glucose levels (Figure 11A), with PIO leading the trend, followed by FC-122, glibenclamide, FC-114, and acarbose. A similar pattern was observed in TAG levels (Figure 11C). On the other hand, FC-114 demonstrated a dramatically increasing trend for insulin levels (Figure 11B) suggesting a secretagogue effect. In fact, the insulin levels produced by FC-114 were greater than those of glibenclamide, which is a secretagogue drug. Interestingly, the FC-114 compound decreased total cholesterol (Figure 11D) and LDL cholesterol (Figure 11F) without affecting HDL levels (Figure 11E). This is an attractive strategy for preventing T2D complications such as ischemic heart disease. When only the following three groups were analyzed (Figure 11G,H), T2D animals without treatment, T2D animals treated with FC-114, and T2D animals treated with FC-122, significant differences were observed (*p* > 0.01). Notably, FC-114 exerts a marked increase in insulin levels and a decrease in total cholesterol levels when compared to untreated T2D animals. Taken together, these results suggest that adding an indole ring to the GA skeleton improves anti-hypercholesterolemic activity and insulin secretion while adding an *N*-phenylpyrazole ring to the GA skeleton improves the anti-hyperglycemic effect. 

The blood glucose-lowering effect and improved insulin levels of FC-114 and FC-122 may be linked to their activity as PTP1B or α-glucosidase inhibitors, according to our in silico and in vitro results. However, it has been reported that GA enhances glucose-stimulated insulin secretion and increases mRNA levels of insulin receptor substrate-2, pancreas duodenum homeobox-1, and glucokinase [85]. In fact, Kalaiarasi et al. reported an increase in insulin levels in STZ-induced diabetic rats treated with 100 mg/kg of GA [31,32]. Additionally, due to the increased insulin secretion, these animals showed an increase in HDL levels and a decrease in glucose, total cholesterol, and cLDL levels in the plasma. Therefore, according to our findings and the shared GA skeleton, it is possible that FC-114 and FC-122 may act also through this mechanism. Despite their significant potency against the PTP1B enzyme, it was expected that both FC-114 and FC-122 would be better anti-diabetic agents in vivo. However, their unexpected effectiveness may be attributed to their high lipophilicity (Log *p* = 7.8 and 6.9, respectively), which may contribute to their high metabolic turnover, low solubility, and poor oral absorption.

The aim of this study was to investigate whether FC-114 and FC-122 have potential anti-diabetic activity in vivo by blocking either PTP1B or α-glucosidase or both. However, the study has certain limitations in establishing a relationship between the anti-diabetic effect of both compounds and the inhibition of these enzymes. Further research is required, such as determining PTP1B and α-glucosidase mRNA levels in the tissues of diabetic rodents treated with the compounds. Additionally, this study was unable to determine if the compounds increase insulin sensitivity, which is a potential effect of PTP1B inhibitors [40]. To confirm this effect, insulin receptor phosphorylation and GLUT4 expression should be measured [86]. It is important to note that the doses of FC-114 and FC-122 used in this study did not significantly impact blood glucose levels, which is a limitation. To address this, it may be necessary to increase the doses to observe significant differences. Furthermore, measuring the expression of PTP1B and GLUT4, as well as insulin phosphorylation, should be considered to confirm the mechanism of the compounds’ pharmacological activity.

## 3. Materials and Methods

### 3.1. General

All chemicals and starting materials were obtained from Sigma-Aldrich (Toluca MEX, Mexico and St. Louis, MO, USA). Reactions were monitored by TLC on 0.2 mm percolated silica gel 60 F254 plates (Sigma-Aldrich) and visualized using irradiation with a UV lamp. Melting points were determined in a BUCHI (model M-565) melting point apparatus. According to IUPAC rules, compounds were named using the automatic name generator tool implemented in ChemDraw Professional 16.0.1.4 software (PerkinElmer, Waltham, MA, USA).

### 3.2. Synthesis

The commercially available GL ammonium salt was used for the synthesis of GA derivatives (Figure 1). GA, FC-114, and FC-122 were synthesized using the procedure described in Ref. [53] with slight modifications. Compounds FC-114 and FC-122 were prepared in a multi-gram scale as described below.

#### 3.2.1. Synthesis of GA

A solution of GL ammonium salt (10 g, 11.9 mmol) in 125 mL of water and 26 mL of conc. HCl was refluxed for 24 h. The resulting precipitate was collected using vacuum filtration and washed with 250 mL of water, yielding 4.8 g (85%) of GA as a brown light solid. This compound was pure enough for the following steps.

#### 3.2.2. Synthesis of 3-Oxo-Glycyrrhethinic Acid (**2**)

This compound was prepared from 4.8 g (10.2 mmol) of GA according to the procedure described by Gao, et al. [87] to give 2 (4.3 g, 90%) of 2 as a white solid.

#### 3.2.3. Synthesis of 3,11-Dioxo-Olean-12-En-30-Oic Acid (**3**)

This compound was prepared from 5.6 g of **2** according to the procedure described by Gao, et al. [87] to give 2 (3.4 g, 61%) of 3 as a white solid.

#### 3.2.4. Synthesis of FC-114

A mixture of ketone **2** (2.35 g, 5 mmol), 4-(trifluoromethyl)phenylhydrazine hydrochloride (1.3 g, 6.1 mmol), and glacial acetic acid (30 mL) was heated at reflux during 2 h. When the mixture reached room temperature, the solid form was collected using filtration and washed with cold glacial acetic acid and water to give 2.6 g (88%) of FC-114 as a light-yellow solid. This compound was pure enough and no additional purification steps were needed. M.p = 261.8 °C; the spectroscopic data agrees with previously reported data [53].

#### 3.2.5. Synthesis of FC-122

To a solution of compound **3** (2.5 g, 5.1 mmol) in dry ethanol (25 mL), 4-tolyl phenylhydrazine hydrochloride (1.6 g, 10.0 mmol) was added. This solution was stirred at 70 °C for 4 h, cooled at room temperature, and the solvent was subsequently evaporated. The crude product was purified using flash chromatography using hexanes/ethyl acetate (from 80:20 to 70:30). Afterwards, the resulting solid was recrystallized from methylene chloride, ethyl acetate, and methanol to give 1.6 g (53%) of FC-122 as an ivory solid. M.p = 235.7 °C; the spectroscopic data agrees with previously reported data [53].

### 3.3. In Vitro Assays

#### 3.3.1. Enzyme Kinetics

The kinetic studies for PTP1B and α-glucosidase were carried out under the same conditions described in the references [53,60,61].

#### 3.3.2. Cytotoxic Effect on Human Foreskin Fibroblasts

In a 96-well plate, 7000 cells per well of Human Foreskin fibroblasts (HFF-1) were cultured. Compounds FC-114, FC-122, and GA were dissolved in dimethyl sulfoxide to give a final concentration of 10 mM. The 50% cytotoxic concentration (CC_50_) was calculated as follows: the cells were incubated with the compounds at concentrations of 0.78, 1.56, 3.11, 6.25, 12.5, 25, 50, and 100 µM for 48 h at 37 °C in a 5% of CO_2_. Untreated cells were also included as a negative control. At the end of the incubation period, the cell contents were precipitated with 50 µL of a 50% aqueous solution of trichloroacetic acid (TCA) at 4 °C for 1 h. The supernatant was discarded and washed 5 times with distilled water. Afterward, the plates were allowed to dry at room temperature. The cell contents were incubated with 100 µL of a 0.04% sulforhodamine B (SRB) solution for 1 h at room temperature. The excess SRB was removed by washing with a 1% aqueous acetic acid solution. To improve the dissolution of the colored complex, 100 µL of Tris buffer pH 10.5 was added to the plate and then read at 490 nm in an Epoch (BioTek^®^) microplate reader. With the data obtained, the viability of treated cells was calculated by comparing the control without treatment. CC_50_ value was determined by employing BioStat Pro 5.9.8 software (AnalystSoft Inc., Brandon, FL, USA).

### 3.4. In Silico Studies

Ligands acarbose, VOG, and GA were retrieved from the Protein Data Bank (PDB) (http://www.rcsb.org/) (accessed on 10 January 2023) (PDB ID: PRD_90010, VOG and CBW, respectively). FC-114 and FC-122 ligands were constructed employing the YASARA structure [70] (version 22.9.24) by systematically modifying the GA’s structure. First, the protonation state of all compounds was fixed, assuming a pH = 7.4, and the 3D geometry was optimized using the PM3 Hamiltonian method as implemented in the YASARA structure package. The three-dimensional crystal structure of α-glucosidase from *R. Obeum* was retrieved from the PDB (http://www.rcsb.org/) (accessed on 10 January 2023) entry 6c9z. Using the YASARA structure, first, the water molecules were removed from the macromolecule. Afterward, their geometry was minimized by employing the YASARA2 forcefield by running the *em_clean* macro. After removing the co-crystallized ligand, the minimized structure (saved as *.pdb format) was used to dock each ligand employing the CB-Dock2 server (https://cadd.labshare.cn/cb-dock2/php/index.php (accessed on 12 January 2023)) [65,66], Autodock 4.2 (The Scripps Research Institute, La Jolla, CA, USA) [67], and GOLD (The Cambridge Crystallographic Data Centre, Cambridge, UK) version 2023.1.0 [68]. 

MD simulations of ligand-protein complexes were performed using the YASARA structure. The results from docking and MD simulations were visualized using PyMOL (The PyMOL Molecular Graphics System, Version 2.0 Schrödinger, LLC) and the Protein-Ligand Interaction Profiler server [88,89], whereas 2D-interaction diagrams were produced with Discovery Studio visualizer 2021 (Dassault Systèmes, San Diego, CA, USA) [90]. The protocol for docking and MD simulations studies was as follows.

#### 3.4.1. Molecular Docking

The CB-Dock2 server (https://cadd.labshare.cn/cb-dock2) (accessed on 12 January 2023) was used for blind molecular docking, while site-specific molecular docking was performed through Autodock 4.2 and GOLD. The graphical interface AutodockTools 1.7.1 [67] suite was used to prepare and analyze the docking simulations. Hydrogen atoms were added to the macromolecules, and Gasteiger-Marsili charges were assigned to the atoms in the protein and ligands. Both protein and ligands were exported as *.pdbqt files. Docking simulations using Autodock 4.2 at the catalytic site of α-glucosidase (using the optimized 6c9z macromolecule) were performed with a grid box size of 60 Å × 60 Å × 90 Å with a spacing of 0.345 Å and coordinate x = −9.814, y = −7.641, and z = 38.544. For blind docking simulations, the minimized structure of α-glucosidase and FC-122 ligand was submitted to the CB-Dock2 server. The best docking conformation with the best binding energy was selected, and the protein-ligand complex was downloaded. Afterward, the ligand was removed from the protein complex, and a site-specific docking simulation was performed at the preferred binding site of FC-122 using a grid box size of 60 Å × 60 Å × 60 Å with a spacing of 0.345 Å and coordinates x = −7.90, y = −8.40, and z = 39.40. The search for VOG, acarbose, and FC-122 was carried out with the Lamarckian Genetic Algorithm. In total, 100 GA runs with a maximum number of 25,000,000 evaluations, a mutation rate of 0.02, and an initial population of 150 conformers were covered. Finally, each ligand with the best cluster size and the lowest binding energy was selected for further analysis.

The resulting ligand complexes obtained from Autodock were exported to GOLD. Using the GOLD wizard, the proteins were prepared by adding hydrogens and extracting the ligands, which were further docked at the catalytic site or allosteric site within a 6 Å radius sphere and were carried out using the following parameters: 100 genetic algorithm runs and 125,000 operations. CHEMPLP fitness was chosen as the main scoring function, whereas GoldScore fitness was chosen as the re-scoring function. The docking simulations were ranked according to the value of the CHEMPLP and GoldScore fitness function.

Using the LigRMSD web server (https://ligrmsd.appsbio.utalca.cl/) (accessed on 12 January 2023), the docking protocol for each software was validated by comparing the RMSD value of the co-crystal structure of VOG and its docked structure [91].

#### 3.4.2. MD Simulations

The ligand-protein complexes were submitted to MD simulations with YASARA structure. The simulations started by running the *md_run* macro, which included an optimization of the hydrogen bonding network to increase the solute stability, and a pKa prediction to fine-tune the protonation states of protein residues at the chosen pH of 7.4. NaCl ions were added with a physiological concentration of 0.9%, with either Na or Cl excess to neutralize the cell. After the steepest descent and simulated annealing minimizations to remove clashes, the simulation was run for 200 nanoseconds using the AMBER14 force field [71] for the solute, GAFF2 [92] and AM1BCC [93] for ligands, and TIP3P for water. The cutoff was 8 Å for Van der Waals forces (the default used by AMBER [94]. No cutoff was applied to electrostatic forces (using the Particle Mesh Ewald algorithm, [95]). The equations of motion were integrated with a multiple-time step of 1.25 fs for bonded interactions and 2.5 fs for non-bonded interactions at a temperature of 310 K and a pressure of 1 atm (NPT ensemble). After the inspection of the solute RMSD as a function of simulation time, the first 100 picoseconds were considered equilibration time and excluded from further analysis. The binding energy study using the MM-PBSA method was performed by running the *md_analyzebindingenergy* macro, which was previously modified with PBS method at a temperature of 310 K.

### 3.5. In Vivo Studies

#### 3.5.1. Animals

Six female (three for each compound: **FC-114** and **FC-122**) (220 ± 10 g, for the acute toxicity assay) and forty male (215 ± 20 g, for the pharmacologic study) albino Wistar rats were acquired from the Isolation and Animal Facility Unit, Facultad de Estudios Superiores Cuautitlán (FESC), Universidad Nacional Autónoma de México (UNAM). They were kept in standard polypropylene boxes under controlled temperature (24 ± 4 °C) on a 12 h light/dark cycle and provided conventional food (formula 5001, Rat Chow) and purified water ad libitum [13]. The groups for the diabetes model (*n* = 6) were as follows: STZ vehicle (control, group 1), diabetic animals only administered with the vehicle of the treatments (group 2), diabetic animals treated with glibenclamide (group 3), FC-114 (group 4), FC-122 (group 5), PIO (group 6), or acarbose (group 7). At the end of the experiment, the animals were taken to the humanitarian endpoint by the intraperitoneal (ip) administration of 75 mg/kg of sodium pentobarbital to perform a necropsy [96] in the bioterium of FESC, UNAM. During the necropsy, blood serum was extracted to quantify glucose, triacylglycerides, cholesterol, and insulin. The study was conducted based on the guidelines of the Declaration of Helsinki and approved by the Institutional Review Board: Internal Committee for the Care and Use of Experimentation Animals (CICUAE, according to the initials in Spanish, C23_02 on March 2023) of the FESC UNAM (Mexico). It complies with the Mexican norm for this matter (NOM-062-ZOO-1999, Technical Specifications for the Production, Care, and Use of Laboratory Animals, SAGARPA), as well as the Guide for the Care and Use of Laboratory Animals of the National Research Council and National Institutes of Health (NIH Publications No. 8023, revised 1978).

#### 3.5.2. Blood Sample Collection and Processing

To obtain serum from each rat under anesthesia, a 10 mL blood sample was taken via cardiac puncture of the exposed heart. After being left at room temperature for 10 min, the blood samples were centrifuged at 2500 rpm (1200× *g*) for 10 min to obtain the serum [83,97]. The resulting serum was then frozen at −70 °C until needed. Prior to usage, the serum samples were thawed at room temperature.

#### 3.5.3. Evaluation of Acute Oral Toxicity

The female Wistar rats (*n* = 3) were used to obtain the AOT based on the procedures of Guide Number 425 of the Organization for Economic Cooperation and Development (OECD) [98]. The rats were closely monitored for 48 h after receiving oral doses of FC-114 or FC-122. Rat 1 received 175 mg/kg, rat 2 received 550 mg/kg, and rat 3 received 2000 mg/kg. They were then observed for 14 days until a necropsy to check for any toxic effects [13,98]. 

#### 3.5.4. Rat Model of Diabetes Induced by Streptozotocin

After a week of acclimation, 35 male Wistar rats were given a single intraperitoneal dose of STZ (Sigma Chemical Co. Toluca, Estado de México, Mexico) at a dosage of 45 mg/kg of body weight, after fasting for 12 h overnight [75,77] with free access to water. To prevent initial drug-induced hypoglycemia mortality [32], these animals were given a 10% sucrose solution 24 h after administration. STZ was dissolved in 0.1 M of sodium citrate buffer (pH 4.5) and administered in a final volume of 5 mL/kg [13,99]. A week after STZ injection, diabetes in streptozotocin rats was confirmed by measuring fasting blood glucose (by glucose oxidase method) [32]. Animals with a blood glucose (tail vein) above 126 mg/dL were considered diabetic and used for the experiment to create groups 2, 3, 4, 5, 6, and 7 [1,13]. Blood glucose levels were measured using a glucometer (Accu-Chek Active, Roche, Mannheim, Germany) and reactive strips (Accu-Chek Active Glucose test strips, Roche). The presence of hyperglycemia and insulin level confirmed a model very similar to T2D in humans [73]. For the pharmacological study, the final number of animals was 35 male rats, with 3 rats being sacrificed during the treatment stage. One was administered with FC-114 and two with acarbose. The first rat was damaged during oral glucose administration in the OGTT at week 5, and the other rats had gastrointestinal complications in week 7.

#### 3.5.5. Administration of Treatments

The diabetic animals in group 2 (untreated) were given 4% *v*/*v* aqueous EtOH solution (vehicle of treatments) in a volume of 6.5 mL/kg body weight. The other groups (3–7) received different treatments for 14 days (weeks 5 to 7). Group 3 was given glibenclamide at 0.6 mg/kg/day, group 4 received FC-114 at 51.3 mg/kg/day, group 5 received FC-122 at 49.1 mg/kg/day, group 6 received PIO at 30 mg/kg/day, and group 7 received acarbose at 54.3 mg/kg/day in a vehicle suspension [85,97]. All of the treatments were at equimolar doses of PIO and adjusted to the weekly weight of each rat. The glibenclamide tablet (5 mg) was crushed in 70 mL of an EtOH solution (4% *v*/*v*) and the vehicle of treatments [73,77] was given to each animal in 0.071 mg/mL of suspension. The commercial PIO tablet (30 mg) was crushed in 7.5 mL of vehicle [73,77] and given to each animal in 4.0 mg/mL of suspension. The FC-114 and FC-122 suspensions were created at 6.2 and 6.0 mg/mL, respectively. The commercial acarbose tablet (50 mg) was crushed in 7.5 mL of the vehicle and given to each animal in a suspension of 6.6 mg/mL. Each suspension was used within 2 days of its preparation, and the administration volume was near to 2.5 mL for each rat.

#### 3.5.6. The Oral Glucose Tolerance Test

The OGTT was carried out in weeks 3, 4, 5, 6, and 7. Prior to the OGTT, all animal groups fasted overnight for eight hours with free access to water. Each rat was weighed and then received a 2 g/kg glucose overload orally and blood samples were taken from the tail vein [13,97]. Glucose levels were measured using a glucometer (Accu-Chek Active, Roche, Mannheim, Germany) at 0, 30, 60, 90, 120, and 180 min after glucose administration [73,100]. The AUC of the OGTT was calculated using the blood glucose concentration over time [86,96] and the GraphPad Prism 8.0.1 software package (GraphPad Software; Science Inc., San Diego, CA, USA) was used to compare the effectiveness of the treatments.

#### 3.5.7. Blood Determinations

The blood determinations were carried out using different kits: glucose (1001191, Spinreact, Girona, Spain, 1045), TAG (41032, Spinreact, LIQ557), insulin (RAB09A-1EA-KC, Sigma, 11092020), total cholesterol (41020, Spinreact, LIQ384), and HDL cholesterol (1001095, Spinreact, H166). Mathematically, the c-LDL cholesterol was determined by subtracting the HDL fraction from the total cholesterol.

#### 3.5.8. Statistical Analysis 

The data were expressed as the mean ± standard error of the mean (M ± SEM) [75,97] and analyzed by one-way analysis of variance (ANOVA), then compared with the Bonferroni post hoc test. Significant differences were considered as 0.05 > *p* > 0.01 (*, significant), 0.01 > *p* > 0.001 (**, very significant), 0.001 > *p* > 0.0001 (***, extremely significant), and *p* < 0.0001 (****, extremely significant). The animals of the vehicle group presented differences from all other groups: α**, very significant, α*** and α****, extremely significant.

## 4. Conclusions

We prepared two GA derivatives, FC-114 and FC-122, in a gram scale starting from GL, a readily available and inexpensive raw material extracted from licorice roots of *G. glabra*. In vitro studies showed that FC-122 is a dual PTP1B/α-glucosidase uncompetitive/mixed-type inhibitor. Docking and MD simulations on α-glucosidase revealed that FC-122 binds to the adjacent region of the catalytic site. Conversely, the in vivo studies on diabetic rats showed that this compound reduces food intake and glucose levels during an oral glucose tolerance test. Meanwhile, FC-114 produced lesser pharmacological results but improved insulin levels and decreased total cholesterol without affecting HDL cholesterol. Further research is needed to establish the mechanism of action and proper dosage to achieve an adequate reduction in blood glucose levels. We plan to assess these compounds in a rat model of T2D using a high-fat diet to investigate the anti-hyperglycemic and anti-lipidemic activity of FC-114 and FC-122 more thoroughly. Finally, we look forward to preparing water-soluble FC-114 or FC-122 derivatives to increase their oral absorption. 

## 5. Patents

The patent application MX/a/2022/004731 is owned by Francisco Cortés-Benítez, Martín González-Andrade, Jaime Pérez-Villanueva, Francisco Palacios-Espinosa, and Félix Matadamas-Martínez.

## Data Availability

The data has been added in the Appendix A section.

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
