# Peer review of "Anti-Diabetic Activity of Glycyrrhetinic Acid Derivatives FC-114 and FC-122: Scale-Up, In Silico, In Vitro, and In Vivo Studies"

_ijms, 2023, doi:10.3390/ijms241612812_

Round 1
Reviewer 1 Report
Samuel Álvarez–Almazán et al., synthesized the derivative compounds of Glycyrrhetinic acid and studied and compared the anti-diabetic activity of FC-114 and FC-122 with other reference drug molecules. The study is novel, and the authors presented experiments of in-silico, in-vitro analysis to characterize the effect of newly synthesized compounds on their substrate enzymes. Also, the authors performed in-vivo experiments on T2D model rats to characterize the anti-diabetic nature of the compounds.
The study has limitations, and the manuscript requires improvements in several areas.
1) The introductory part is clear and well-established for the subsequent results and discussion section. The manuscripts' results and discussion content is not focused on the study's aims, explaining unimportant observations and making irrelevant comparisons. Authors should focus on the aims of their study and limit their content to explaining and discussing their essential observations.
A few examples are,
Throughout the manuscript, the authors compared the molecular changes between vehicle or control condition and T2D+ treatment conditions (mostly reference drugs, including glibenclamide, PIO, and acarbose). Instead, their focus should be comparing the T2D condition and T2D+ treatment, precisely on FC114 and FC122.
The paragraph starting from lane 421 or another paragraph starting lane 446, the authors talk about every observation they made in Figure 9 and point out irrelevant comparisons. Instead, the authors should prioritize their findings and limit their explanation to the focus of the study.
In Figures 8B and 8C, the authors did not find/demonstrate any statistical comparisons between T2D and T2D+ drug treatments, which is essential in the context of the paper. On the other hand, the authors compare the values of food and water intake between the condition of the vehicle and T2D+ treatment, which is the least interesting.
2) Authors often explain their experimental observations and fail to discuss/interpret the findings. For example, the authors demonstrate and explain their findings that drug treatments, particularly at 7 weeks in T2D animals, reduce their ability to consume food and water but never interpret or discuss possible mechanisms/reasons for their observations.
3) Surprisingly, the manuscript data on acarbose treatment of T2D animal models show increased serum glucose and OGTT levels measured at different time points in contrast to several earlier reports. The authors fail to discuss the discrepancies, and further, authors prioritize comparing their observations between T2D+ acarbose and T2D+FC114 or FC122 ( figure 9).
4) The discrepancy between the data presented in the figures and the content explained in the results section is evident in several instances.
For example, in Section 2.6 of the results and discussion, a) Authors claim that FC-114 treatment in T2D animal models reduced food consumption from the second week of the treatment. Whereas the graph presented in Figure 8B demonstrates that food consumption for the T2D+FC-114 condition looks similar to the T2D condition until 6 weeks of the treatment. Also, according to Figure 8b, the authors did not plot/demonstrate food consumption value at 2 weeks of FC-114 treatment.
b) Lane 386-389, based on data presented in Figure 9A, the sentence in lane 386-389 is incorrect. Authors claim that glibenclamide and acarbose are highly effective in reducing plasma glucose levels when compared to other treatments. Figure 9A shows acarbose is the least potential, and glibenclamide did not show any impact after 4 weeks of treatment.
c) Lane 367-368, the Authors mention that PIO and glibenclamide treatment reduced food consumption after 7 days of drug administration. Graph 8b shows that changes are observed at 7 weeks of drug treatment.
d) Lane 370-372 lacks clarity. The authors should rewrite the sentence.
5) Toxicity studies: The experimental evidence to conclude that newly synthesized compounds are not toxic is minimal in the study. The authors did not present any bar chart comparisons of Invitro toxicity studies on HEF-1 cells.
N/A
Author Response
Sincerest thanks for your comments on our manuscript. We sincerely apologize for the great time it has taken us to respond to these comments. Nevertheless, we have done our very best to enrich, clarify and complete the work herein and hope that the revised version of the manuscript can still be considered for publication in IJMS.
All changes in the manuscript are marked by using the “Track Changes” function. The point-by-point responses to the comments of the reviewers are listed below.
1) The introductory part is clear and well-established for the subsequent results and discussion section. The manuscripts' results and discussion content is not focused on the study's aims, explaining unimportant observations and making irrelevant comparisons. Authors should focus on the aims of their study and limit their content to explaining and discussing their essential observations.
A few examples are,
Throughout the manuscript, the authors compared the molecular changes between vehicle or control condition and T2D+ treatment conditions (mostly reference drugs, including glibenclamide, PIO, and acarbose). Instead, their focus should be comparing the T2D condition and T2D+ treatment, precisely on FC114 and FC122.
Response:
We thank reviewer for the suggestion and irrelevant comparisons have been limited and the text now focuses on the findings of the new treatments FC-114 and FC-122. Lines 345, 366, 372, 381, 392, 398, 428, and 458 were removed from the manuscript. In addition, Figure 11G and 11H were added to highlight the activity of FC-114.
The paragraph starting from lane 421 or another paragraph starting lane 446, the authors talk about every observation they made in Figure 9 and point out irrelevant comparisons. Instead, the authors should prioritize their findings and limit their explanation to the focus of the study.
Response: The authors appreciate the comment. Irrelevant comparisons were avoided.
In Figures 8B and 8C, the authors did not find/demonstrate any statistical comparisons between T2D and T2D+ drug treatments, which is essential in the context of the paper. On the other hand, the authors compare the values of food and water intake between the condition of the vehicle and T2D+ treatment, which is the least interesting.
Response: The justification for this is indicated in the line 368, as follows: “The consumption of water and food (figures 9B and 9C) was measured per cage of animals, so it was not possible to obtain the error bars.”
Responding to the other point: The comparative analysis between the animals of the vehicle group (healthy) and the other groups of animals was avoided. The lines 366, 372, 458 were removed from the manuscript.
2) Authors often explain their experimental observations and fail to discuss/interpret the findings. For example, the authors demonstrate and explain their findings that drug treatments, particularly at 7 weeks in T2D animals, reduce their ability to consume food and water but never interpret or discuss possible mechanisms/reasons for their observations.
Response: The following text was added to indicate the possible mechanisms for reducing water and food consumption in line 388, as follows:
“The treatments that produce a decrease in hyperglycemia levels in diabetic patients/animals produce a decrease in fasting glucose, which in turn produces molecular variations such as less thirst (due to the osmotic capacity of glucose) and less presence of the hunger stimulus (due to insulin sensitization and the ability of glucose to enter tissues sensitive to this hormone) [10,15,83]. For this reason, there is no feedback to generate the hunger stimulus and the animals stop eating a lot.”
3) Surprisingly, the manuscript data on acarbose treatment of T2D animal models show increased serum glucose and OGTT levels measured at different time points in contrast to several earlier reports. The authors fail to discuss the discrepancies, and further, authors prioritize comparing their observations between T2D+ acarbose and T2D+FC114 or FC122 ( figure 9).
Response: We thank for the comment. It was not indicated that the treatment with acarbose increased glucose levels, perhaps there was a lack of clarity in the wording to explain that the administration of the treatments started at week 5. To clarify this, some modifications were made in the lines 351 and 353 to mention the following:
“This model generated with 45 mg/kg of STZ (administered at week 1) did not cause weight loss (Figure 8A), and apparently, the FC–114 (p > 0.01) and PIO (p > 0.001) treatments (administered for 14 days: week 5 to 7 after STZ injection) caused a difference in weight compared to animals without treatment (T2D).”
In the OGTT test, the figures 10B and 10C (9b and 9c for the previous manuscript) were removed to avoid misunderstandings.
4) The discrepancy between the data presented in the figures and the content explained in the results section is evident in several instances.
For example, in Section 2.6 of the results and discussion, a) Authors claim that FC-114 treatment in T2D animal models reduced food consumption from the second week of the treatment. Whereas the graph presented in Figure 8B demonstrates that food consumption for the T2D+FC-114 condition looks similar to the T2D condition until 6 weeks of the treatment. Also, according to Figure 8b, the authors did not plot/demonstrate food consumption value at 2 weeks of FC-114 treatment.
Response: The paragraph was rewritten to highlight that the treatments were administered during weeks 5 to 7, considering the administration of the STZ in week 1. On the other hand, the food intake values of the animals with the treatments FC-114, FC-122 and PIO are very similar, so they overlap making them appear to be absent. The changes are in lines 351, 353, 370, and 375 as follows:
“This model generated with 45 mg/kg of STZ (administered at week 1) did not cause weight loss (Figure 8A), and apparently, the FC–114 (p > 0.01) and PIO (p > 0.001) treatments (administered for 14 days: week 5 to 7 after STZ injection) caused a difference in weight compared to animals without treatment (T2D).”
“The food intake values of the animals with the treatments FC-114, FC-122 and PIO are very similar, so they overlap making them appear to be absent.”
Paragraph in line 375 was reprased
b) Lane 386-389, based on data presented in Figure 9A, the sentence in lane 386-389 is incorrect. Authors claim that glibenclamide and acarbose are highly effective in reducing plasma glucose levels when compared to other treatments. Figure 9A shows acarbose is the least potential, and glibenclamide did not show any impact after 4 weeks of treatment.
Response: Since the treatments started at week 5 and ended at week 7, the weekly changes in fasting glucose suggest that acarbose and glibenclamide have a greater effect than the rest of the treatments after two weeks of administration.
c) Lane 367-368, the Authors mention that PIO and glibenclamide treatment reduced food consumption after 7 days of drug administration. Graph 8b shows that changes are observed at 7 weeks of drug treatment.
Response: c) In the same way as the previous comment, the text of lines 351, 353, 371, and 375 were inserted to clarify this confusion.
d) Lane 370-372 lacks clarity. The authors should rewrite the sentence.
Response: The comparison with the vehicle group (healthy) animals was avoided to focus only on the novel compounds. The paragraph of those lines has been rephrased (lines 375 to 384).
5) Toxicity studies: The experimental evidence to conclude that newly synthesized compounds are not toxic is minimal in the study. The authors did not present any bar chart comparisons of In vitro toxicity studies on HEF-1 cells.
Response: We are agreeing with this comment. A bar chart with the in vitro citoxicity results is now reported in Figure 6.
Reviewer 2 Report
In this research, Álvarez–Almazán and colleagues investigate the anti-diabetic potential of Glycyrrhetinic acid derivatives, namely FC–114 and FC–122. The authors determined that FC–122 has a strong inhibitory effect on PTP1B and α-glucosidase enzymes, which play a significant role in managing Type 2 diabetes (T2D). FC–114 also displayed noteworthy PTP1B inhibitory activity. The mechanism by which FC–122 inhibits α-glucosidase was revealed through molecular docking and molecular dynamics studies. Experiments using diabetic rats induced by Streptozotocin (STZ) demonstrated that FC–122 and FC–114 could enhance glucose, insulin, and cholesterol levels. Notably, FC–122 was observed to decrease food consumption and glucose levels in the GTT, while FC–114 significantly boosted insulin levels and lowered total cholesterol levels without impacting HDL cholesterol.
There are a couple of major concerns that need to be addressed:
- The study lacks experimental validation of mechanisms by which GA derivates function to exert their anti-diabetic effects. Experiments showing mechanisms are necessary.
- STZ model doesn't recapitulate T2D pathogenesis as it damages beta cells. A high-fat diet model would be more suitable, and studies in this model are warranted.
Author Response
Sincerest thanks for your comments on our manuscript. We sincerely apologize for the great time it has taken us to respond to these comments. Nevertheless, we have done our very best to enrich, clarify and complete the work herein and hope that the revised version of the manuscript can still be considered for publication in IJMS.
All changes in the manuscript are marked by using the “Track Changes” function. The point-by-point responses to the comments of the reviewers are listed below.
There are a couple of major concerns that need to be addressed:
- The study lacks experimental validation of mechanisms by which GA derivates function to exert their anti-diabetic effects. Experiments showing mechanisms are necessary.
Response: The authors appreciate the reviewer's comments. This study was developed around the hypothesis of the main inhibition of the PTP1B and α-glucosidase enzymes, two pharmacological targets whose participation has been described in T2D. Much remains to be known about PTP1B. For this reason, the study of the enzymatic inhibition on this target was carried out and the hypothesis of the in vitro results on this inhibition was raised. Determining the mechanism of action of the new compounds was not the objective of this study. The next stage is to assess how these compounds achieve their pharmacological actions.
- STZ model doesn't recapitulate T2D pathogenesis as it damages beta cells. A high-fat diet model would be more suitable, and studies in this model are warranted.
Response: The authors greatly appreciate the reviewer´s suggestions. The STZ model causes damage to the beta cells of the pancreas by increasing oxidative stress and this limits insulin production without it disappearing. This molecular event has also been observed in T2D patients, where the same excess glucose maintains this oxidative stress, increasing damage to the pancreas. In addition, it is one of the most reported models and closest to the signs and symptoms present in diabetic patients because it focuses on the partial damage of beta cells. Changes to highlight this are found on line 347 as follows: “It is known that the STZ produces a rat model with severe weight loss [32], signs and symptoms [13,73-75,76,96] like the events in diabetic patients.”
In addition, we wrote a sentence in Line 736, where we look forward to assess these compounds using the suggested model.
Round 2
Reviewer 1 Report
The authors made several necessary changes. The current state of the manuscript looks much improved for readers to follow it.
Since treatments of different drugs started after five weeks of inducing diabetes, for the readers' clarity, it is better to indicate the same on every graph with an arrow mark at week 5 ( to point out the fact that drug treatments started at five weeks, otherwise it is easy to assume that treatments might have begun along with STZ treatment).
I will also encourage the authors to double-check grammatical mistakes in the manuscript.
Author Response
We would like to express our deepest gratitude for your feedback on our manuscript. We sincerely apologize for the significant delay in responding to your comments. We have made every effort to enhance, clarify, and finalize the content, and we hope that the revised version of the manuscript will still be deemed suitable for publication in IJMS.
The modifications made to the manuscript are indicated through the use of the "Track Changes" feature. Below, you will find a detailed reply to each reviewer's comment.
1) Since treatments of different drugs started after five weeks of inducing diabetes, for the readers' clarity, it is better to indicate the same on every graph with an arrow mark at week 5 ( to point out the fact that drug treatments started at five weeks, otherwise it is easy to assume that treatments might have begun along with STZ treatment). I will also encourage the authors to double-check grammatical mistakes in the manuscript.
Response: The graphs have been updated with markings that indicate the administration of treatments to prevent any confusion. Furthermore, we conducted a thorough grammar check on the manuscript.
Reviewer 2 Report
Although the authors assert that it was not the objective of their study, without mechanism there remains a significant gap in the manuscript.
Author Response
We would like to express our deepest gratitude for your feedback on our manuscript. We sincerely apologize for the significant delay in responding to your comments. We have made every effort to enhance, clarify, and finalize the content, and we hope that the revised version of the manuscript will still be deemed suitable for publication in IJMS.
The modifications made to the manuscript are indicated through the use of the "Track Changes" feature. Below, you will find a detailed reply to each reviewer's comment.
1) Although the authors assert that it was not the objective of their study, without mechanism there remains a significant gap in the manuscript.
Response: The study focused on the effects of derivatives on the enzymes PTP1B and α-glucosidase, which are targeted by glycyrrhetinic acid (GA). Inhibition kinetics were determined, and it was found that GA derivatives have inhibitory capacity. This suggests that the compounds may work by inhibiting either PTP1B, α-glucosidase, or both. These findings were supported by in vitro results. We are aware that we could use Western blot to determine the expression of both enzymes. However, this would require us to redo the experiments in vivo, a process that would take at least two months. Additionally, we lack the necessary resources and infrastructure to conduct such experiments, including the required reagents.
Round 3
Reviewer 2 Report
The issue of lack of resources is understandable. I would recommend the authors to include a detailed paragraph adding the limitations of this study so that it is clear for the reader and other other researchers about the caveats of the study.
Author Response
We want to express our deepest gratitude for your feedback on our manuscript. We sincerely apologize for the significant delay in responding to your comments. We have made every effort to enhance, clarify, and finalize the content, and we hope that the revised version of the manuscript will still be deemed suitable for publication in IJMS.
The modifications made to the manuscript are indicated through the use of the "Track Changes" feature.
Comment: The issue of lack of resources is understandable. I would recommend the authors to include a detailed paragraph adding the limitations of this study so that it is clear for the reader and other researchers about the caveats of the study.
Response: We wrote a paragraph in line 500 explaining our study's limitations. The paragraph is as follows:
"The aim of this study was to investigate whether FC-114 and FC-122 have potential antidiabetic activity in vivo by blocking either PTP1B or α-glucosidase or both. However, the study has certain limitations in establishing a relationship between the antidiabetic effect of both compounds and the inhibition of these enzymes. Further research is required, such as determining PTP1B and α-glucosidase mRNA levels in the tissues of diabetic rodents treated with the compounds. Additionally, this study was unable to determine if the compounds increase insulin sensitivity, which is a potential effect of PTP1B inhibitors [40]. To confirm this effect, insulin receptor phosphorylation and GLUT4 expression should be measured [100]. It's important to note that the doses of FC–114 and FC–122 used in this study did not significantly impact blood glucose levels, which is a limitation. To address this, it may be necessary to increase the doses to observe significant differences. Furthermore, measuring the expression of PTP1B and GLUT4, as well as insulin phosphorylation, should be considered to confirm the mechanism of the compounds' pharmacological activity."